# Surfactant delivery via thin catheter in preterm infants: A systematic review and meta-analysis

**Telford Y. Yeung**[1,2,3]*, **Qi Zhou**[4], **H. Godze Kanmaz Kutman**[5], **Aakash Pandita**[6], **Eleni Philippopoulos**[7], **Bonny Jasani**[8,9]

1 Section of Neonatology, Windsor Regional Hospital, Windsor, Canada, 2 Department of Biomedical Sciences, University of Windsor, Windsor, Canada, 3 Schulich School of Medicine Windsor Campus, Western University, Windsor, Canada, 4 Deparment of Neonatology, Children's Hospital of Fudan University, Shanghai, China, 5 Department of Pediatrics, Ankara Bilkent City Hospital, Ankara, Turkey, 6 Division of Neonatology, Medanta Hospital, Lucknow, India, 7 Mount Sinai Hospital, Toronto, Canada, 8 Division of Neonatology, Hospital for Sick Children, Toronto, Canada, 9 Department of Pediatrics, University of Toronto, Toronto, Canada

* telford.yeung@wrh.on.ca

**Data Availability Statement:** All relevant data are within the manuscript and its Supporting Information files.

## Abstract

### Objective

Surfactant administration via a thin catheter (STC) is an alternative to surfactant administration post endotracheal intubation in preterm infants with respiratory distress syndrome (RDS); however, the benefits particularly in infants <29 weeks' gestation and the neurodevelopmental outcomes remain unclear. Thus, our objective was to systematically review and meta-analyze the efficacy and safety of STC compared to intubation for surfactant or nasal continuous positive airway pressure (nCPAP) in preterm infants with RDS.

### Methods

Medical databases were searched until December 2022 for randomized controlled trials (RCTs) assessing STC compared to controls that included intubation for surfactant or nCPAP in preterm infants with RDS. The primary outcome was bronchopulmonary dysplasia (BPD) at 36 weeks gestation in survivors. Subgroup analysis was conducted comparing STC to controls in infants < 29 weeks' gestation. The Cochrane risk of bias (ROB) tool was used and certainty of evidence (CoE) was rated according to GRADE.

### Results

Twenty-six RCTs of 3349 preterm infants, in which half of the studies had low risk of bias, were included. STC decreased the risk of BPD in survivors compared to controls (17 RCTs; N = 2408; relative risk (RR) = 0.66; 95% confidence interval (CI) 0.51 to 0.85; number needed to treat for an additional beneficial outcome (NNTB) = 13; CoE: moderate). In infants < 29 weeks' gestation, STC significantly reduced the risk of BPD compared to controls (6 RCTs, N = 980; RR 0.63; 95% CI 0.47 to 0.85; NNTB = 8; CoE: moderate).

**Funding:** The authors received no specific funding for this work.

**Competing interests:** The authors have declared that no competing interests exist.

**Abbreviations:** BPD, bronchopulmonary dysplasia; CI, confidence interval; CoE, certainty of evidence; nCPAP, nasal continuous positive airway pressure; INSURE, intubation for surfactant administration followed by extubation; IVH, intraventricular hemorrhage; MV, invasive mechanical ventilation; NNTB, number needed to treat for an additional beneficial outcome; RCT, randomized controlled trial; RDS, respiratory distress syndrome; ROB, risk of bias; RR, relative risk; STC, surfactant via thin catheter.

## Conclusions

Compared to controls, STC may be a more efficacious and safe method of surfactant delivery in preterm infants with RDS, including infants < 29 weeks' gestation.

## Introduction

Respiratory distress syndrome (RDS) is the most common respiratory condition in preterm infants that is best characterized as a primary deficiency in surfactant [1, 2]. As such, surfactant therapy (ST) where indicated and respiratory support via non-invasive or invasive mechanical ventilation (MV) are the standards of care [1–4]. In preterm infants, ST reduces the need for respiratory support and the rates of pneumothorax and mortality [3, 5]. Furthermore, dosing [3, 6], timing [7, 8], type [9, 10] and the method of surfactant delivery are additional considerations that can affect morbidity and mortality related to RDS [3, 4]. INSURE (intubation for surfactant followed by extubation), an alternative approach to care, can decrease air leaks, bronchopulmonary dysplasia (BPD) and the need for MV when compared to intubation with ongoing MV [11]. However, its superiority to nasal continuous positive airway pressure (nCPAP) in decreasing the risk of BPD or death remains uncertain [12].

To mitigate the risk of BPD associated with MV, an alternative strategy for surfactant delivery using a thin catheter (STC) has been described [3, 13, 14]. A prior meta-analysis of randomized controlled trials (RCTs) demonstrated a reduction in the risk of BPD for preterm infants with STC compared to intubation for surfactant therapy [13]. Moreover, STC reduced the need for MV but had no effect on in-hospital mortality; however, short and long-term outcomes for infants less than 29 weeks' birth gestation were not described in this review.

As the incidence of BPD continues to increase in extremely preterm neonates [15], advances in neonatal respiratory care are still needed [15]. Although STC appears effective and safe for preterm infants [13], its effect on morbidity and mortality in neonates < 29 weeks' gestation is unclear. Hence, our objective was to systematically review and meta-analyze the efficacy and safety of STC compared to INSURE, intubation for surfactant and MV with delayed extubation, or nCPAP for preterm infants with RDS, including in infants < 29 weeks' gestation.

## Methods

We reported the results of our review using the Cochrane Neonatal Review Group guidelines for conducting a systematic review [16] and the Preferred Reporting Items for Systematic Reviews and Meta-analyses (PRISMA) guidelines [17]. Ethics approval was not obtained for this review. This review was registered with Open Science Framework (OSF) [osf.io/tcdwz].

### Eligibility criteria

**Types of studies.** RCTs which compared the efficacy or safety of STC to INSURE, intubation for surfactant and MV with delayed extubation, or nCPAP alone in preterm infants with RDS were included. Case studies, cross-over studies, retrospective studies, prospective observational studies, systematic reviews, narrative reviews, editorials and animal-based studies were read in detail to identify eligible studies but were excluded from the analysis.

**Participants.** Preterm (less than $37^{+0}$ weeks birth gestation) infants with RDS or at risk of RDS and requiring STC compared to INSURE, intubation for surfactant and MV with delayed

extubation or nCPAP alone were included. RDS was usually diagnosed by respiratory distress within several hours after birth and/or radiographic features consistent with RDS.

**Interventions and control.** The intervention group comprised of STC that involved passing a thin catheter (enteral feeding tube or stiff vascular catheter) with direct or video laryngoscopy beyond the cords for surfactant delivery while on non-invasive ventilation. The control group included infants treated with INSURE or intubation and MV with delayed extubation or stabilization with nCPAP alone.

**Outcomes.** The primary outcome was bronchopulmonary dysplasia (BPD) at 36 weeks' gestation in survivors [18]. The secondary outcomes were the composite outcome of BPD or death, mortality prior to discharge, need for intubation within 72 hours of birth, need for more than one dose of surfactant, adverse events such as pneumothorax and intraventricular hemorrhage (IVH), and neurodevelopmental outcomes at 18–24 months corrected age.

## Search strategy

Medline, Excerpta Medica database (Embase), Cumulative Index of Nursing and Allied Health Literature (CINAHL), Cochrane Central Register of Controlled Trials (CENTRAL) databases and the China National Knowledge Infrastructure Database (CNKI) were searched until December 31, 2022. EP conducted independent searches of the various medical databases. No language restrictions were applied to the searches. We also searched the first 400 hits in Google Scholar for articles that may not have been indexed in standard medical databases.

## Study selection

Abstracts of the studies were read in completion by TY and QZ to identify potential studies for inclusion in the analysis. Full texts of the articles were then obtained and read by the authors. For multiple publications with the same data, this was considered as one study to avoid duplication of data. Differences were resolved via consensus by involving another author (BJ).

## Data extraction

TY and QZ independently extracted the data using a data collection form. Study design and outcomes were reviewed and verified by TY and QZ. Discrepancies in data extraction were resolved by discussion until a consensus was achieved. Translation for journal articles in Chinese was performed by QZ. The authors of several studies were contacted for clarification and to obtain individual patient data in infants < 29 weeks' gestation.

## Assessment for risk of bias (ROB)

Assessment of the ROB followed the Cochrane Collaboration ROB Assessment Tool for RCTs [16]. TY and QZ independently assessed ROB for the following domains: random number generation, allocation concealment, blinding of intervention and outcome assessors, completeness of follow up, selectivity of reporting and other potential biases. ROB was assigned as low, unclear and high risk based on the Cochrane Collaboration guidelines. Judgement regarding the risk of bias for blinding of patients and study personnel was excluded because blinding in this domain was not possible. Study personnel could not be blinded to the procedure due to the nature of the intervention, as these procedures were different. Studies were considered low risk if rated low risk in the remaining domains with a maximum of one domain rated unclear risk. Studies were deemed moderate risk if they had at least two domains with unclear risk and no domains in the high-risk category. High risk studies had at least one domain rated high risk.

## Subgroup analysis

We compared the efficacy and safety of STC to controls in infants less than 29 weeks' gestation. We also contacted and requested data from corresponding authors for studies that included infants < 29 weeks' gestation.

## Statistical analysis

The meta-analysis was performed with the RevMan 5.3.5 software as part of the Cochrane Collaboration (Nordic Cochrane Centre, Copenhagen, Denmark). Forest plots were calculated using weighted scores and a random effects model (REM, Mantel Haenszel method). We employed REM to account for variations in methodology of surfactant delivery and gestational age. Statistical heterogeneity was assessed with a $\chi^2$ test and the $I^2$ statistic. A p-value of $< 0.1$ for the $\chi^2$ statistic indicated significant heterogeneity. For the $I^2$ statistic, we used the following thresholds: 0–40%: might not be important; 30–60%: may represent moderate heterogeneity; 50–90%: may represent substantial heterogeneity; 75–100%: considerable heterogeneity [16, 19]. For studies that presented data as median and interquartile range, we estimated the mean and standard deviation using the minimum and maximum values as well as the interquartile ranges [20]. To combine means and standard deviations, we used calculations provided by the Cochrane Handbook [16]. Effect size was reported as relative risk (RR) and associated 95% confidence interval (CI) or mean difference (MD) and 95% CI as appropriate. Publication bias was assessed by a funnel plot [16].

## Summary of findings

Key information about the study including certainty of evidence (CoE), details of the intervention and summary of outcome data were included in the summary of findings table according to the Grading of Recommendations, Assessment, Development and Evaluation (GRADE) guidelines. Grading of evidence was performed with the online tool GradePro GDT [21].

## Results

### Study selection

The literature search identified 766 records in Embase, 569 records in Medline, 424 in Pubmed, 246 in CINAHL, 357 results in CENTRAL, 75 from CNKI and 303 from Google Scholar. Full texts of thirty-one RCTs were reviewed in depth. Three studies were excluded due to duplication of data [22–24]. One RCT (published abstract of the trial protocol), was also excluded because neither the methodology nor results could be assessed [25]. One study was a non-randomized prospective trial [26]. The remaining twenty-six RCTs met our inclusion criteria. The results of this search are presented in **Fig 1**.

### Characteristics of studies/trials

Infants between 23 to 37 weeks gestation were included in these RCTs. Of the twenty-six included RCTs [27–52], twelve studies enrolled infants < 29 weeks' gestation [27–30, 32, 33, 37, 38, 41, 43, 44, 48]. We obtained data from 7 studies for infants < 29 weeks' gestation for subgroup analysis [27, 28, 32, 38, 41, 43, 48]. Neurodevelopmental outcomes were reported in two studies [53, 54]. Five studies used the stiff vascular catheter [31, 36, 38, 39, 48] and twenty studies used a feeding tube with or without forceps [27–30, 32, 34, 35, 37, 40–47, 49–52]; one study did not specify the technique of administration [33]. Two studies compared STC to intubation for surfactant with MV [32, 34], two studies used nCPAP as a control group [27, 48],

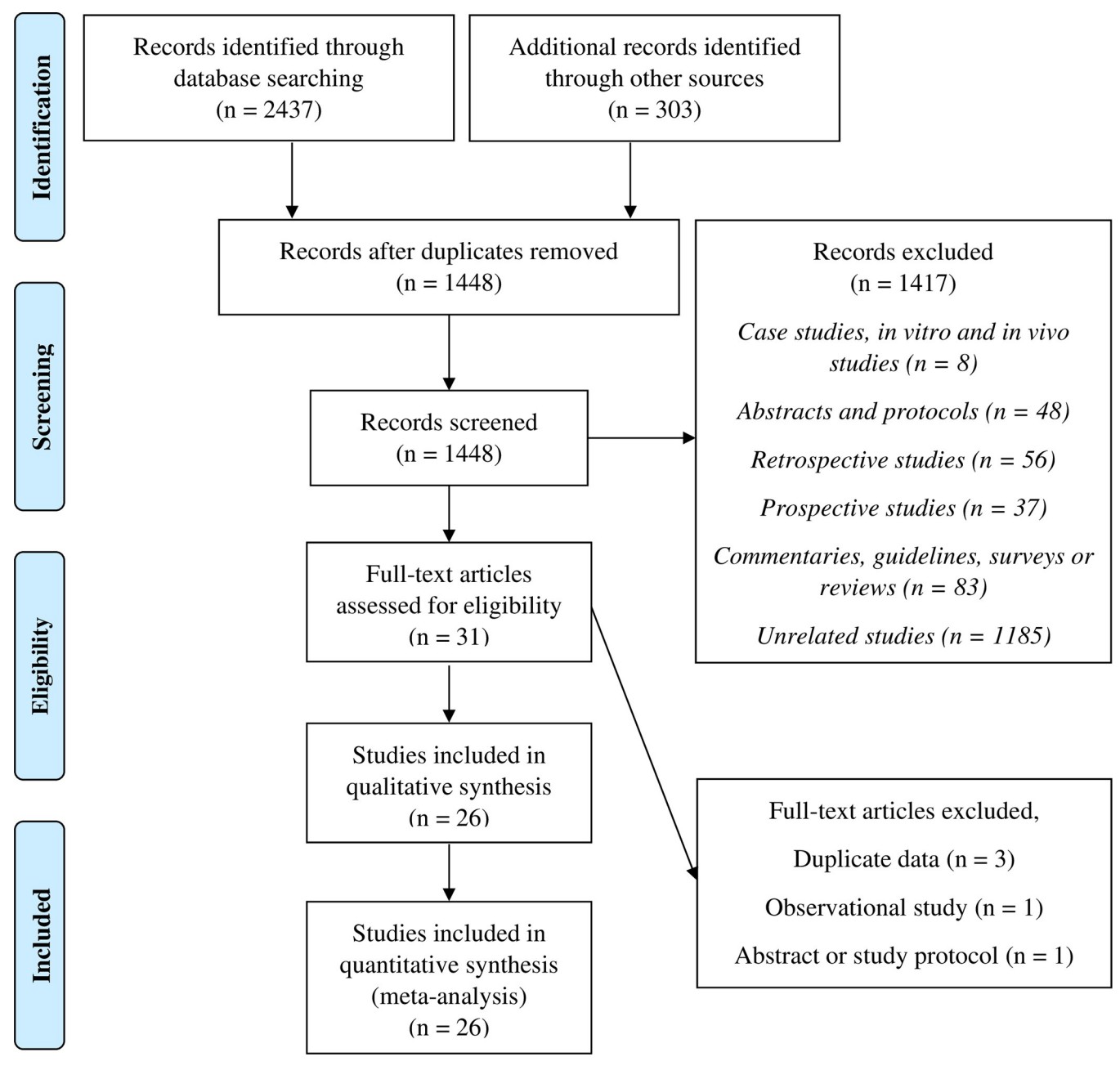

**Fig 1. Search results from medical databases.**

while the remaining 22 studies used INSURE as their control group. A summary of the included trials is presented in **S1 Table**.

## Risk of bias

Using the Cochrane ROB tool, we found that 50% of studies (13/26) had unclear ROB for allocation concealment and 42% of studies (11/26) had unclear ROB for random sequence generation. The details of the ROB assessment are documented in **Fig 2**.

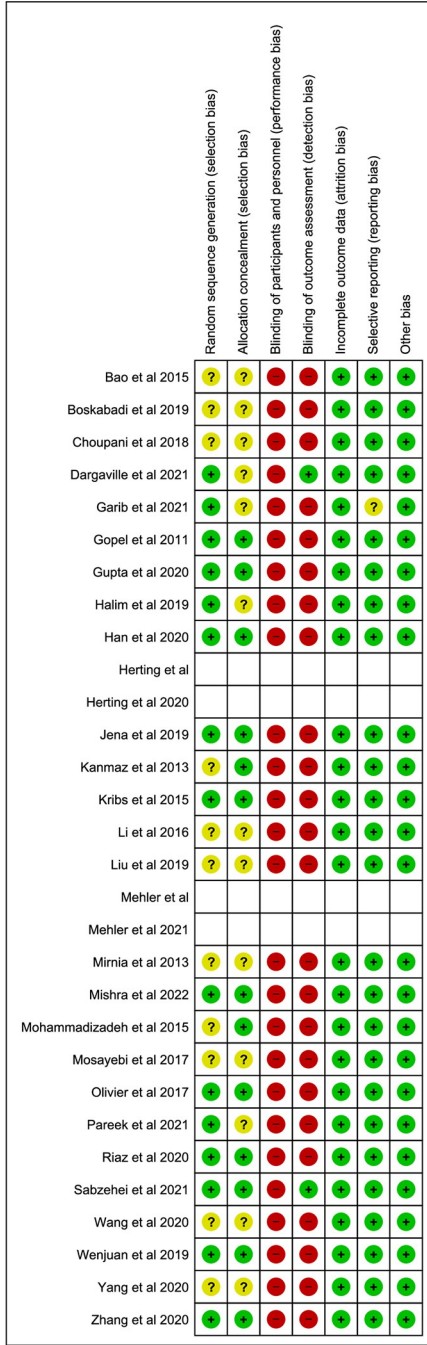

**Fig 2. Risk of bias assessment.**

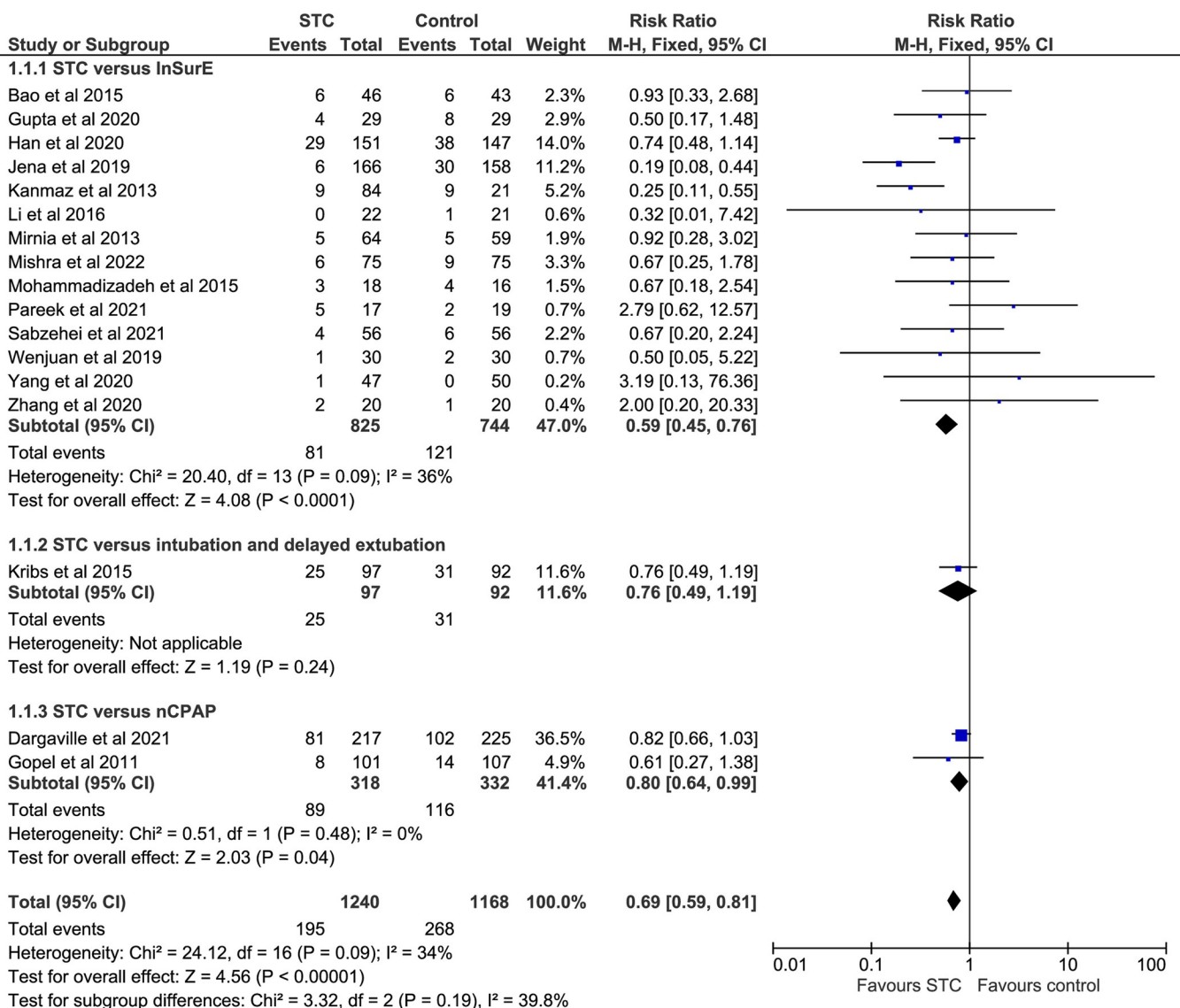

**Fig 3. Forest plot of STC versus controls for BPD at 36 weeks among survivors.**

## Primary outcome

The rates of BPD were decreased for survivors who were treated with STC compared to controls (17 RCTs, N = 2408; Relative risk (RR) 0.66; 95% confidence interval (CI) 0.51 to 0.85; NNTB = 13; CoE: moderate;) [Fig 3].

## Secondary outcomes

**Composite outcome of BPD at 36 weeks or death.** Compared to the control group, there was a significant decrease in the composite outcome of BPD or death with the STC group (22 RCTs, N = 3045, RR 0.71; 95% CI 0.60 to 0.84; NNTB = 12; CoE: moderate) [**Fig 4**].

**In-hospital mortality.** There was no significant decrease for in-hospital mortality with STC versus the control group. (17 RCTs, N = 2606, RR 0.78; 95% CI 0.58 to 1.05; CoE: moderate) [**Table 1**]. While a reduction in mortality was observed with STC versus INSURE; this

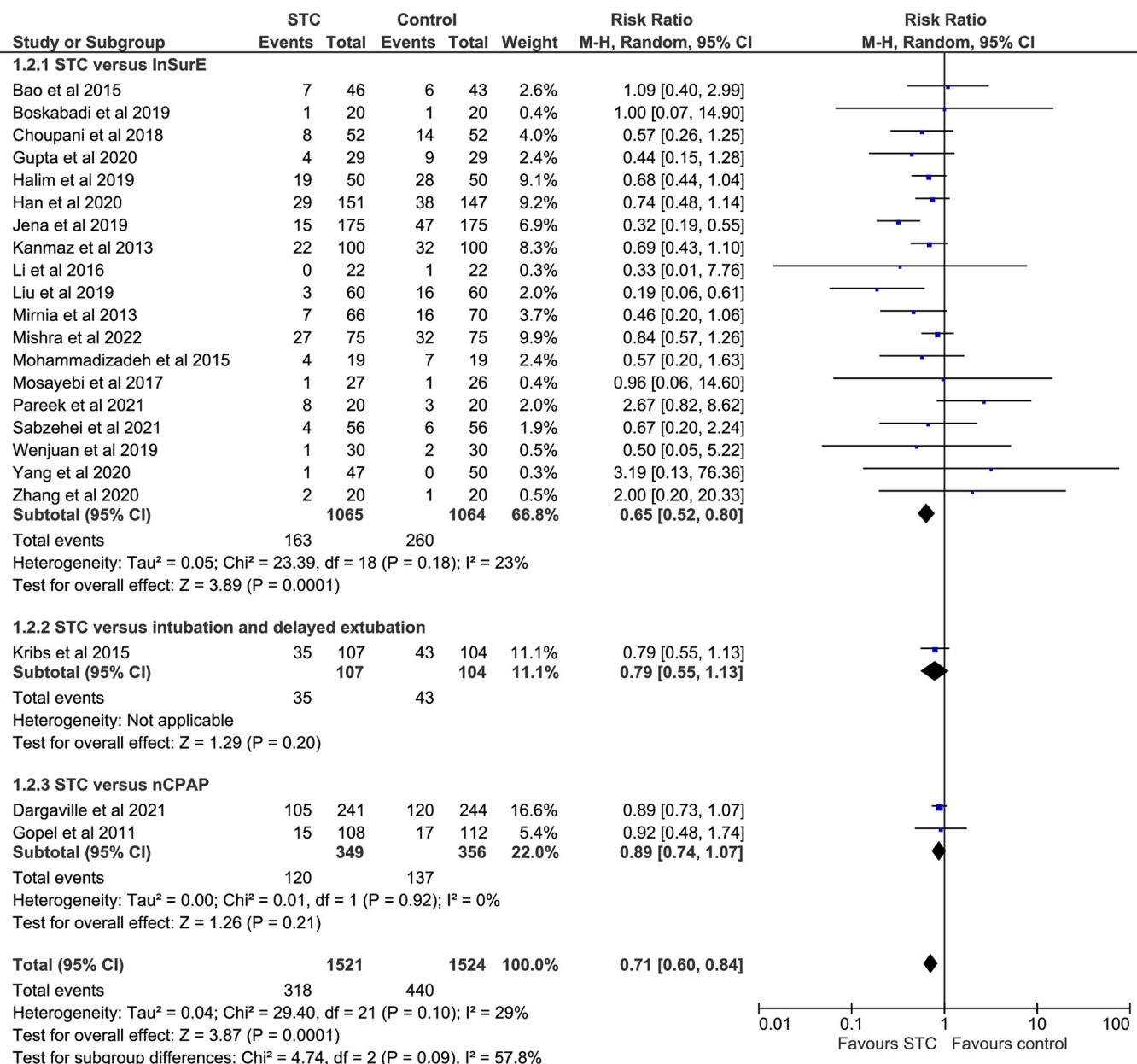

**Fig 4. Forest plot of STC versus controls for BPD or death.**

reduction was offset by no mortality benefit in STC versus nCPAP or delayed extubation [Table 1].

**Need for intubation within 72 hours of birth.** The need for intubation within 72 hours of birth was decreased with STC compared to the control group (18 RCTs, N = 2369; RR 0.65; 95% CI 0.55 to 0.77; CoE: moderate) [Table 1].

**Adverse events.** Pneumothorax occurred less frequently with STC compared to the control group (14 RCTs, N = 1895; RR 0.45; 95% CI 0.26 to 0.78; CoE: moderate) [Table 1].

**Long-term outcomes (neurodevelopmental and pulmonary).** Neurodevelopmental outcomes were reported in a secondary analysis of two of the RCTs from Germany [53, 54]. Pooled data showed no differences between the two groups for the Bayley II mental

**Table 1. Summary of findings for surfactant via thin catheter (STC) in preterm infants with RDS.**

| Outcome | Number of participants | Relative risk | Anticipated absolute effects | | CoE |
|---|---|---|---|---|---|
| | | | Risk with STC | Risk with Control | |
| **BPD at 36 weeks among survivors** | 2408 (17 RCTs) | 0.66 (95% CI 0.51–0.85) | 151 per 1000 | 229 per 1000 | Moderate[a] |
| STC versus INSURE | 1569 (14 RCTs) | 0.60 (95% CI 0.41–0.89) | 98 per 1000 | 163 per 1000 | Moderate[a] |
| STC versus intubation and delayed extubation | 189 (1 RCT) | 0.76 (95% CI 0.49–1.19) | 256 per 1000 | 337 per 1000 | Low[a,b] |
| STC versus nCPAP | 650 (2 RCTs) | 0.81 (95% CI 0.65–1.00) | 283 per 1000 | 349 per 1000 | Moderate[a] |
| **BPD or death** | 3045 (22 RCTs) | 0.71 (95% CI 0.60–0.84) | 205 per 1000 | 289 per 1000 | Moderate[a] |
| STC versus INSURE | 2129 (19 RCTs) | 0.65 (95% CI 0.52–0.80) | 159 per 1000 | 244 per 1000 | Moderate[a] |
| STC versus intubation and delayed extubation | 211 (1 RCT) | 0.79 (95% CI 0.55–1.13) | 327 per 1000 | 413 per 1000 | Low[a,b] |
| STC versus nCPAP | 705 (2 RCTs) | 0.89 (95% CI 0.74–1.07) | 343 per 1000 | 385 per 1000 | Moderate[a] |
| **Mortality** | 2606 (20 RCTs) | 0.78 (95% CI 0.58–1.05) | 112 per 1000 | 143 per 1000 | Moderate[a] |
| STC versus INSURE | 1690 (17 RCTs) | 0.68 (95% CI 0.47–0.96) | 122 per 1000 | 179 per 1000 | Moderate[a] |
| STC versus intubation and delayed extubation | 211 (1 RCT) | 0.81 (95% CI 0.37–1.79) | 93 per 1000 | 115 per 1000 | Low[a,b,c] |
| STC versus nCPAP | 705 (2 RCTs) | 1.31 (95% CI 0.79–2.19) | 88 per 1000 | 67 per 1000 | Low[a,c] |
| **Need for intubation within 72 hours after birth** | 2369 (18 RCTs) | 0.65 (95% CI 0.55–0.77) | 287 per 1000 | 442 per 1000 | Moderate[a] |
| STC versus INSURE | 1408 (14 RCTs) | 0.72 (95% CI 0.58–0.90) | 231 per 1000 | 321 per 1000 | Moderate[a] |
| STC versus intubation and delayed extubation | 256 (2 RCTs) | 0.53 (95% CI 0.25–1.15) | 301 per 1000 | 568 per 1000 | Low[a,b] |
| STC versus nCPAP | 705 (2 RCTs) | 0.53 (95% CI 0.45–0.62) | 338 per 1000 | 638 per 1000 | Moderate[a] |
| **Need for greater than one dose of surfactant** | 1716 (15 RCTs) | 1.17 (95% CI 0.94–1.44) | 169 per 1000 | 144 per 1000 | Moderate[a] |
| STC versus INSURE | 1671 (14 RCTs) | 1.15 (95% CI 0.92–1.42) | 163 per 1000 | 142 per 1000 | Moderate[a] |
| STC versus intubation and delayed extubation | 45 (1 RCT) | 1.57 (95% CI 0.63–3.96) | 374 per 1000 | 238 per 1000 | Low[a,b,c] |
| **Pneumothorax** | 1895 (14 RCTs) | 0.45 (95% CI 0.26–0.78) | 43 per 1000 | 95 per 1000 | Moderate[a] |
| STC versus INSURE | 1038 (10 RCTs) | 0.68 (95% CI 0.41–1.14) | 48 per 1000 | 71 per 1000 | Moderate[a] |
| STC versus intubation and delayed extubation | 253 (2 RCTs) | 0.39 (95% CI 0.15–0.96) | 47 per 1000 | 121 per 1000 | Low[a,b] |
| STC versus nCPAP | 604 (2 RCTs) | 0.16 (95% CI 0.02–1.41) | 21 per 1000 | 129 per 1000 | Low[a,b,c] |
| **IVH ($\geq$ grade II)** | 2752 (18 RCTs) | 0.78 (95% CI 0.60–1.01) | 69 per 1000 | 88 per 1000 | Moderate[a] |
| STC versus INSURE | 1836 (15 RCTs) | 0.84 (95% CI 0.60–1.18) | 62 per 1000 | 74 per 1000 | Moderate[a] |
| STC versus intubation and delayed extubation | 211 (1 RCT) | 0.46 (95% CI 0.24–0.90) | 102 per 1000 | 221 per 1000 | Low[a,b] |
| STC versus nCPAP | 705 (2 RCTs) | 0.88 (95% CI 0.53–1.46) | 74 per 1000 | 84 per 1000 | Moderate[a] |

a–lack of blinding for participants and outcome assessors

b–Small sample size

c–wide confidence intervals

developmental index (Mean difference (MD) -1.52; 95% CI -11.52 to 8.47; CoE: very low) [**S13 Fig**] and the psychomotor developmental index (MD 4.88; 95% CI -8.23 to 17.98; CoE: very low) [**S14 Fig**] at 24 months of age; but one study showed significantly fewer patients with severe disability in the psychomotor developmental index with STC compared to controls (22% (15/68) versus 42% (29/69), p = 0.012) [54].

**Additional secondary outcomes.**   There were no significant differences between the two groups for other pre-specified outcomes [**Table 1**].

## Subgroup analysis

**BPD at 36 weeks in surviving preterm infants $<$ 29 weeks' gestation.**   The rates of BPD in surviving preterm infants $<$ 29 weeks' gestation treated with STC were significantly decreased compared to the control group (6 RCTs, N = 980; RR 0.63; 95% CI 0.47 to 0.85; NNTB = 8; CoE: moderate) [**S6 Fig**].

Table 2. Summary of findings for STC in infants < 29 weeks' gestation with RDS.

| Outcome | Number of participants | Relative risk or mean difference | Anticipated absolute effects | | CoE |
|---|---|---|---|---|---|
| | | | Risk with STC | Risk with control | |
| **BPD at 36 weeks among survivors** | 980 (6 RCTs) | 0.63 (95% CI 0.47–0.85) | 230 per 1000 | 365 per 1000 | Moderate[a,b] |
| STC versus INSURE | 139 (3 RCTs) | 0.40 (95% CI 0.25–0.65) | 197 per 1000 | 492 per 1000 | Moderate[a] |
| STC versus intubation and delayed extubation | 189 (1 RCT) | 0.76 (95% CI 0.49–1.19) | 256 per 1000 | 337 per 1000 | Low[a,b] |
| STC versus CPAP | 652 (2 RCTs) | 0.81 (95% CI 0.65–1.00) | 281 per 1000 | 347 per 1000 | Moderate[a] |
| **BPD or death** | 1112 (7 RCTs) | 0.81 (95% CI 0.69–0.94) | 350 per 1000 | 432 per 1000 | Moderate[a] |
| STC versus INSURE | 196 (4 RCTs) | 0.69 (95% CI 0.47–1.01) | 430 per 1000 | 624 per 1000 | Moderate[a] |
| STC versus intubation and delayed extubation | 211 (1 RCT) | 0.79 (95% CI 0.55–1.13) | 327 per 1000 | 413 per 1000 | Low[a,b] |
| STC versus CPAP | 705 (2 RCTs) | 0.88 (95% CI 0.73–1.06) | 341 per 1000 | 388 per 1000 | Moderate[a] |
| **Mortality** | 1105 (7 RCTs) | 1.00 (95% CI 0.78–1.27) | 115 per 1000 | 115 per 1000 | Moderate[a] |
| STC versus INSURE | 189 (4 RCTs) | 0.94 (95% CI 0.70–1.25) | 282 per 1000 | 300 per 1000 | Moderate[a] |
| STC versus intubation and delayed extubation | 211 (1 RCT) | 0.81 (95% CI 0.37–1.79) | 93 per 1000 | 115 per 1000 | Low[a,b,d] |
| STC versus CPAP | 705 (2 RCTs) | 1.31 (95% CI 0.79–2.19) | 88 per 1000 | 67 per 1000 | Low[a,d] |
| **Need for intubation within 72 hours after birth** | 1105 (7 RCTs) | 0.60 (95% CI 0.51–0.69) | 352 per 100 | 587 per 1000 | Moderate[a] |
| STC versus INSURE | 189 (4 RCTs) | 0.70 (95% CI 0.49–1.00) | 334 per 1000 | 478 per 1000 | Moderate[a] |
| STC versus intubation and delayed extubation | 211 (1 RCT) | 0.75 (95% CI 0.55–1.02) | 382 per 1000 | 510 per 1000 | Low[a,b] |
| STC versus CPAP | 705 (2 RCTs) | 0.53 (95% CI 0.45–0.62) | 338 per 1000 | 638 per 1000 | Moderate[a] |
| **Need for one additional dose of surfactant** | 189 (4 RCTs) | 1.01 (95% CI 0.61–1.65) | 258 per 1000 | 256 per 1000 | Low[a,b] |
| **Pneumothorax** | 1102 (7 RCTs) | 0.49 (95% CI 0.31–0.77) | 47 per 1000 | 97 per 1000 | Moderate[a] |
| STC versus INSURE | 189 (4 RCTs) | 0.74 (95% CI 0.26–2.11) | 58 per 1000 | 78 per 1000 | Low[a,d] |
| STC versus intubation and delayed extubation | 208 (1 RCT) | 0.38 (95% CI 0.14–1.02) | 48 per 1000 | 126 per 1000 | Low[a,b] |
| STC versus CPAP | 705 (2 RCTs) | 0.46 (95% CI 0.26–0.84) | 43 per 1000 | 93 per 1000 | Moderate[a] |
| **IVH (≥ grade II)** | 1105 (7 RCTs) | 0.72 (95% CI 0.51–1.01) | 94 per 1000 | 131 per 1000 | Moderate[a] |
| STC versus INSURE | 189 (4 RCTs) | 0.77 (95% CI 0.40–1.46) | 163 per 1000 | 211 per 1000 | Moderate[a] |
| STC versus intubation and delayed extubation | 211 (1 RCT) | 0.46 (95% CI 0.24–0.90) | 102 per 1000 | 221 per 1000 | Low[a,b] |
| STC versus CPAP | 705 (2 RCTs) | 0.88 (95% CI 0.53–1.46) | 74 per 1000 | 84 per 1000 | Moderate[a] |
| **Bayley mental developmental index** | 335 (2 RCTs) | -1.52 (95% CI -11.52 to 8.47) | 1.52 lower (11.52 lower to 8.47 higher) | The mean MDI was 0 | Very low[a,b,c] |
| **Bayley psychomotor developmental index** | 316 (2 RCTs) | -4.88 (95% CI -8.23 to 17.98) | 4.88 lower (8.32 lower to 17.98 higher) | The mean MDI was 0 | Very low[a,b,c] |

a–lack of blinding for participants and outcome assessors

b–Small sample size

c–significant heterogeneity ($I^2$ >50%)

d–wide confidence intervals

**Additional outcomes in preterm infants < 29 weeks' gestation.** There was a decrease in the composite outcome of BPD or death (NNTB = 13; CoE: moderate), the need for intubation within 72 hours and the rates of pneumothoraces [Table 2]. However, there was no difference in mortality prior to discharge, the rates of IVH and the need for more than one dose of surfactant between STC and the control group [Table 2].

## Certainty of evidence and summary of findings

The overall CoE for all outcomes varied from very low to moderate. Using GRADE assessment, there was a concern for risk of bias in the domains of blinding of the participants and outcome assessors. Imprecision was present due to wide confidence intervals for mortality in the comparison of STC with intubation followed by delayed extubation or nCPAP and pneumothorax in the comparison of STC and nCPAP. Publication bias was not detected for the primary outcome in this review based on a funnel plot (**S15 Fig**).

## Discussion

In this review, we report that STC decreases the rate of BPD without decreasing mortality in preterm infants in the most comprehensive meta-analysis to date; these findings concur with previous meta-analyses [13, 14, 55, 56]. Herein, we also present low to moderate certainty data from seven studies that demonstrate decreased BPD, pneumothorax and a reduction in the need for MV without increased mortality or IVH in infants born < 29 weeks' gestation who were managed with STC.

This reduction in BPD observed with STC is likely multifactorial; these factors include the avoidance of MV as well as mitigation of atelectotrauma with simultaneous use of nCPAP during surfactant delivery [13, 55]. Although animal studies have not addressed the reduction in BPD, they have demonstrated that animals treated with STC have improvements in oxygenation and lung mechanics that are similar to intubation with surfactant for both preterm and adult animals [57, 58]. Moreover, we demonstrate that the need for additional doses of surfactant to manage RDS is no different between STC and controls [**Table 1**]. These findings suggests that although STC is less invasive, its effects are comparable to surfactant delivery via endotracheal tube.

BPD is not only important in the neonatal period but also in early childhood, as it is associated with neurodevelopmental impairment and worsened respiratory health in young children [59]. In this meta-analysis, limited follow up data were available. Neurodevelopmental data showed mixed results. Although there was no difference in Bayley II scores between STC and controls, Mehler et al., however, observed a decrease in the proportion of infants with severe disability in the psychomotor development index for the STC group in comparison to intubation for surfactant administration [54]. In addition, follow up of longterm respiratory outcomes remain unclear. Forced expiratory volume in one second (FEV1) at 5 years of age was improved, but at 5–9 years of age, there were no differences between STC and intubation for surfactant [53, 60]. The same research group, however, showed a non-significant decrease in bronchitis with STC compared to controls (34% versus 49%, p = 0.06) [53].

To date, many small trials from around the world have compared STC to controls but fewer studies have examined neonates < 29 weeks' gestation, a sub-group at the highest risk for BPD. There are six large trials with 200 or more patients that included infants at high-risk for BPD [27, 28, 32, 38, 43, 48]. We included one additional trial that shared data on infants < 29 weeks' gestation [41]. In addition, two trials compared STC to nCPAP, which reflects the more contemporaneous neonatal practice of initially managing RDS with nCPAP alone [27, 48]. Data on these trials used different approaches for STC: one with a forceps guided catheter placement and the other with a stiff catheter placement. Nevertheless, the two trials showed no decrease in BPD or mortality between STC and nCPAP; but from the perspective of resource allocation, STC reduced that the rates of intubation and pneumothorax, potentially averting the need for a ventilator or a chest tube, respectively.

## Strengths and limitations

Although this meta-analysis is the largest to date with additional data for infants < 29 weeks' gestation and long-term outcomes, this review is not without limitations. First, many of the trials were at high risk of bias from the lack of blinding for both the study personnel and the outcome assessors. While blinding of study personnel is logistically challenging and requires significant resources and coordination, blinding of outcome assessors is achievable and should be expected in future RCTs of STC. Second, the outcome of BPD has several confounding factors. BPD has multiple definitions, and the absence of a standard definition makes comparison across studies challenging. In addition, there is considerable variation in the amount and duration of exposure to mechanical ventilation in the control group, which can contribute to potentially different baseline rates of BPD in controls [61]. Third, the differences in the administration of surfactant, the type of surfactant, the use of sedation, and the varying approaches to laryngoscopy (video or direct) are some additional features that may introduce confounders into the analysis, as these may influence the need for additional intubation attempts [61]. Lastly, in the subgroup analysis, the sample size was relatively small, increasing the risk for imprecision. More importantly, Dargaville et al. observed an increase in mortality for the infants between 25–26 weeks gestation [48]. While this observed increase in mortality may have been related to chance, caution for infants less than 26 weeks' gestation is warranted until further studies can provide more certainty on these findings. In addition, reporting of neurodevelopmental outcomes were limited by the small number of studies and were based on the older Bayley II assessments.

## Future directions

Current data regarding STC for preterm infants show a reduction in BPD and BPD or death, but the long-term impact on neurodevelopment remains unclear. While preliminary follow up analysis of the neurodevelopmental outcomes for this intervention has demonstrated a reduction in severe developmental impairment, neurodevelopmental data from a greater number of studies, assessment using the most modern Bayley tool (Bayley III or higher) and assessment at school age may provide a more reliable measure of the long-term effects of STC. The long-term effects on pulmonary function or rates of lung infections also needs attention. Finally, determining a safe gestational age for STC will be necessary as a recent multicentre trial suggested that infants < 26 weeks' may be exposed to greater harm from this intervention [48].

As STC can reduce endotracheal intubation, adopting this practice may benefit resource limited settings including low and middle income (LMIC) countries. Implementation of new procedures or equipment can incur costs; however, STC can be performed with standard equipment used for routine neonatal care. But, the learning curve to be proficient with STC remains unexplored. Thus, exploring the cost and economic impact of STC in LMIC should be considered in future studies.

## Conclusion

STC may be superior to controls n reducing BPD and BPD or death for spontaneously breathing preterm infants. Moreover, infants less than 29 weeks' gestation treated with STC also benefit from a reduction in BPD. Therefore, adoption of STC into routine clinical practice is reasonable for preterm infants. The long-term effects on neurodevelopment and respiratory health outcomes require more investigation.

## Supporting information

**S1 Checklist. PRISMA 2020 checklist.**
(DOCX)

**S1 Table. Search strategy.**
(DOCX)

**S2 Table. List of included studies.**
(DOCX)

**S3 Table. List of excluded studies.**
(DOCX)

**S1 Fig. Forest plots of STC versus controls for mortality in preterm infants with RDS.**
(TIF)

**S2 Fig. Forest plots of STC versus controls for need for intubation at 72 hours of life in preterm infants with RDS.**
(TIF)

**S3 Fig. Forest plots of STC versus controls for need for an additional dose of surfactant in preterm infants with RDS.**
(TIF)

**S4 Fig. Forest plots of STC versus controls for pneumothorax in preterm infants with RDS.**
(TIF)

**S5 Fig. Forest plots of STC versus controls for intraventricular hemorrhage in preterm infants with RDS.**
(TIF)

**S6 Fig. Forest plot of STC versus controls for BPD in infants < 29 weeks' with RDS.**
(TIF)

**S7 Fig. Forest plot of STC versus controls for BPD or death in infants < 29 weeks' with RDS.**
(TIF)

**S8 Fig. Forest plot of STC versus controls for mortality in infants < 29 weeks' with RDS.**
(TIF)

**S9 Fig. Forest plot of STC versus INSURE and intubation for surfactant with delayed extubation for need for intubation in infants < 29 weeks' with RDS.**
(TIF)

**S10 Fig. Forest plot of STC versus INSURE for an additional dose of surfactant in infants < 29 weeks' with RDS.**
(TIF)

**S11 Fig. Forest plot of STC versus controls for pneumothorax in infants < 29 weeks' with RDS.**
(TIF)

**S12 Fig. Forest plot of STC versus controls for intraventricular hemorrhage in infants < 29 weeks' with RDS.**
(TIF)

**S13 Fig. Forest plots of STC versus controls for Bayley MDI in preterm infants with RDS.**
(TIF)

**S14 Fig. Forest plots of STC versus controls for Bayley PDI in preterm infants with RDS.**
(TIF)

**S15 Fig. Funnel plot of STC versus controls for publication bias.**
(TIF)

## Acknowledgments

We would like to thank Dr. Alia Halim (PAF Hospital Islamabad, Pakistan) and Dr. Tongyan Han (Children's Hospital of Chongqing Medical University, China) for sharing data from their published work.

## Author Contributions

**Conceptualization:** Telford Y. Yeung, Eleni Philippopoulos, Bonny Jasani.

**Data curation:** Telford Y. Yeung, Qi Zhou, H. Godze Kanmaz Kutman, Aakash Pandita, Eleni Philippopoulos, Bonny Jasani.

**Formal analysis:** Telford Y. Yeung, Qi Zhou, Bonny Jasani.

**Investigation:** Telford Y. Yeung.

**Methodology:** Telford Y. Yeung, Qi Zhou, H. Godze Kanmaz Kutman, Aakash Pandita, Eleni Philippopoulos.

**Project administration:** Telford Y. Yeung.

**Supervision:** Bonny Jasani.

**Writing – original draft:** Telford Y. Yeung.

**Writing – review & editing:** Qi Zhou, H. Godze Kanmaz Kutman, Aakash Pandita, Bonny Jasani.

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
