## [Decision Letter · Decision Letter 0]

14 Mar 2023

PONE-D-23-03856Surfactant Delivery via Thin Catheter in Preterm Infants: A Systematic Review and Meta-AnalysisPLOS ONE

Dear Dr. Yeung,

Thank you for submitting your manuscript to PLOS ONE. After careful consideration, we feel that it has merit but does not fully meet PLOS ONE’s publication criteria as it currently stands. Therefore, we invite you to submit a revised version of the manuscript that addresses the points raised during the review process. Please submit your revised manuscript by Apr 28 2023 11:59PM. If you will need more time than this to complete your revisions, please reply to this message or contact the journal office at plosone@plos.org. Please include the following items when submitting your revised manuscript:A rebuttal letter that responds to each point raised by the academic editor and reviewer(s). You should upload this letter as a separate file labeled 'Response to Reviewers'.A marked-up copy of your manuscript that highlights changes made to the original version. You should upload this as a separate file labeled 'Revised Manuscript with Track Changes'.An unmarked version of your revised paper without tracked changes. You should upload this as a separate file labeled 'Manuscript'.

We look forward to receiving your revised manuscript.

Kind regards,

Harald Ehrhardt

Academic Editor

PLOS ONE

Journal Requirements:

2. We note that this manuscript is a systematic review or meta-analysis; our author guidelines therefore require that you use PRISMA guidance to help improve reporting quality of this type of study. Please upload copies of the completed PRISMA checklist as Supporting Information with a file name “PRISMA checklist”.

Additional Editor Comments:

All four reviewers found merrit in your meta-analysis. I kindly ask you to carefully address all their valuable comments that will help to clarify outstanding points and questions when submitting a revised version to Plos One.

Reviewers' comments:

Reviewer's Responses to Questions

**Comments to the Author**

1. Is the manuscript technically sound, and do the data support the conclusions?

Reviewer #1: Partly

Reviewer #2: Yes

Reviewer #3: No

Reviewer #4: Yes

2. Has the statistical analysis been performed appropriately and rigorously? 

Reviewer #1: No

Reviewer #2: Yes

Reviewer #3: Yes

Reviewer #4: Yes

3. Have the authors made all data underlying the findings in their manuscript fully available?

Reviewer #1: No

Reviewer #2: No

Reviewer #3: Yes

Reviewer #4: Yes

4. Is the manuscript presented in an intelligible fashion and written in standard English?

Reviewer #1: Yes

Reviewer #2: Yes

Reviewer #3: No

Reviewer #4: Yes

5. Review Comments to the Author

Reviewer #1: This manuscript reports meta-analysis results about the efficacy and safety of surfactant via thin catheter administration (STC) compared to intubation for surfactant or nasal continuous positive airway pressure (nCPAP) in preterm infants with RDS, including infants < 29 weeks’ gestation. I have below comments,

It is important to provide data from each individual study that used for meta-analysis. Besides Fig. 3 and 4, please also provide forest plots with data from each individual study for the rest outcomes listed in Table 1A and B.

Please provide funnel plot to show the result of testing publication bias.

Line 215-255 for long-term outcomes, the reported results are different from the data shown in S1 and S2 Figures.

From S1 and S2 Figures, the heterogeneities are significant among studies, where fixed-effect models are not suitable. Random-effect meta-analysis model should be applied. Please correct.

Reviewer #2: The authors deserve credit for their interesting article. A few items need their attention:

1. Line 4: please indicate what #3 stands for

2. Keywords should be written in full

3. Line 46: InSurE should be written as INSURE throughout the manuscript

4. Lines 75 and 77: please explain NNTB abbreviation

5. Line 139: EP stands for the librarian Eleni Philippopoulos, why is she not a co-author?

6. Lines 160-163: difficult to interpret, please rephrase

7. From line 219 onwards the authors frequently use " versus controls". The findings are easier to understand if the authors indicate which support the controls received.

8. The supplemental information contains figures as eps files. I could not open these files, so I could not check whether all data underlying the findings in the manuscript were fully available.

Reviewer #3: Re: Surfactant Delivery via Thin Catheter in Preterm Infants: A Systematic Review and

Meta-Analysis by Telford Y. Yeung et al

I applaud authors for taking up this meta-analyses. This review includes studies published until Dec 2022. This review will be a good addition to the existing literature. However, the meta-analysis has few major deficiencies that should be addressed before it can be published

Abstract:

Methods: The primary outcome of the study is BPD in survivors- the most important outcome of interest for neonatologists is the composite death or BPD and has been the primary outcome of 2 important studies comparing surfactant via thin catheter vs ETT/nCPAP.

Results: Include RR for death/BPD and mortality

Introduction:

L85-86: As such, surfactant replacement therapy (SRT) where indicated and respiratory support via non-invasive or invasive mechanical ventilation (MV) are the standards of care (1-4)- Non invasive mechanical ventilation cannot be called as standard of care – CPAP and NIV can be grouped together with the term non invasive respiratory support.

SRT as abbreviation is quite confusing as authors also use STC. Surfactant replacement therapy can be just called as surfactant therapy without abbreviation. ‘Replacement ’is not an appropriate as the infants have surfactant deficiency and there is nothing to replace.

L 90-91: ‘InSurE (intubation for surfactant followed by extubation), the current standard of care, can decrease air leaks, bronchopulmonary dysplasia (BPD) and the need for MV when compared to ongoing MV’ INSURE cannot be called as current standard of care, it is rather an approach to care

96-97: A prior meta-analysis of randomized controlled trials (RCTs) demonstrated a reduction in the risk of BPD for preterm infants with STC compared to intubation for surfactant therapy (13): death or BPD ?

101-102: Although STC appears effective and safe for preterm infants (13), its effect on morbidity and mortality in neonates < 29 weeks’ gestation is unclear: This sentence is misleading. Although it is true that the previous meta-analysis did not conduct subgroup analysis using GA stratification, almost all (14/16) included infants born in that GA. This statement should be revised.

122: RDS was usually diagnosed by respiratory distress within several hours after birth and radiographic features consistent with RDS: Many studies did not use chest radiograph prior to STC- it could be modified to say infants with RDS or at risk of RDS

L 304-311: It is unclear what the authors are trying to say.

336-341:” Lastly, in the subgroup analysis, the sample size was relatively small, increasing the risk for imprecision. More importantly, Dargaville et al. observed an increase in mortality for the infants between 25 - 26 weeks gestation (48). This finding suggests that STC may not be safe for infants less than 26 weeks ‘gestation, but further studies are required to understand the factors contributing to this increase in mortality” This statement contradicts the findings of the current meta-analysis which shows no difference in the mortality in the entire group or subgroup.

350-352: Finally, determining a safe gestational age for STC will be necessary as a recent mullti center trial suggested that infants < 26 weeks’ may be exposed to greater harm from this intervention : the study findings from Dargaville et al show no difference in mortality between the study groups but a higher mortality in the subgroup analysis. This is probably a chance finding. This study was stopped due to pandemic prior to completing study enrollment. Study also had imbalances between study groups. The discussion related to this study needs to happen (if authors were to include it in the discussion) in the complete context of the study findings and in the light of the current meta-analysis.

Reviewer #4: Authors have done a commendable job in presenting a technically sound & properly worded as well as written manuscript. Relevant data is made available. Data extraction and presentation is quite good.

6. PLOS authors have the option to publish the peer review history of their article (what does this mean?). If published, this will include your full peer review and any attached files.

Reviewer #1: No

Reviewer #2: No

Reviewer #3: No

Reviewer #4: **Yes: **Prof L S Deshmukh

---

## [Author Response · Author response to Decision Letter 0]

27 Mar 2023

Please refer to the appended response to reviewers document in the resubmission for a detailed point by point response to reviewer comments.

---

## [Decision Letter · Decision Letter 1]

10 Apr 2023

Surfactant Delivery via Thin Catheter in Preterm Infants: A Systematic Review and Meta-Analysis

PONE-D-23-03856R1

Dear Dr. Yeung,

We’re pleased to inform you that your manuscript has been judged scientifically suitable for publication and will be formally accepted for publication once it meets all outstanding technical requirements.

Within one week, you’ll receive an e-mail detailing the required amendments.

Please revise the following language errors within the manuscript at this time point:

L124-125: Preterm (less than 37+0 weeks birth gestation) infants with RDS or at risk of RDS and requiring ST via STC… suggest remove ST as an acronym

L143: EP conducted independent searches of the various medical databases…. what does EP stand for? I assume this is an author's initial but EP also stands for many other things. Please clarify

NNB is not consistently noted - it is important to be show NNB for death or BPD in main group and subgroup.

When these have been addressed, you’ll receive a formal acceptance letter and your manuscript will be scheduled for publication.

Kind regards,

Harald Ehrhardt

Academic Editor

PLOS ONE

Additional Editor Comments (optional):

Reviewers' comments:

Reviewer's Responses to Questions

**Comments to the Author**

1. If the authors have adequately addressed your comments raised in a previous round of review and you feel that this manuscript is now acceptable for publication, you may indicate that here to bypass the “Comments to the Author” section, enter your conflict of interest statement in the “Confidential to Editor” section, and submit your "Accept" recommendation.

Reviewer #1: All comments have been addressed

Reviewer #2: All comments have been addressed

Reviewer #3: All comments have been addressed

2. Is the manuscript technically sound, and do the data support the conclusions?

Reviewer #1: (No Response)

Reviewer #2: Yes

Reviewer #3: Yes

3. Has the statistical analysis been performed appropriately and rigorously? 

Reviewer #1: (No Response)

Reviewer #2: Yes

Reviewer #3: Yes

4. Have the authors made all data underlying the findings in their manuscript fully available?

Reviewer #1: (No Response)

Reviewer #2: Yes

Reviewer #3: Yes

5. Is the manuscript presented in an intelligible fashion and written in standard English?

Reviewer #1: (No Response)

Reviewer #2: Yes

Reviewer #3: Yes

6. Review Comments to the Author

Reviewer #1: (No Response)

Reviewer #2: The authors have appropriately addressed the questions raised by the reviewers and made sure that their data are available to the reader.

Reviewer #3: Re: Surfactant Delivery via Thin Catheter in Preterm Infants: A Systematic Review and

Meta-Analysis by Yeung et al

I congratulate authors for taking up this meta-analysis. Authors have addressed the concerns adequately.

Few minor points to be corrected.

L124-125: Preterm (less than 37+0 weeks birth gestation) infants with RDS or at risk of RDS and requiring ST via STC… suggest remove ST as an acronym

L143: EP conducted independent searches of the various medical databases…. what does EP stand for? I assume this is an author's initial but EP also stands for many other things. Please clarify

NNB is not consistently noted - it is important to be show NNB for death or BPD in main group and subgroup.

7. PLOS authors have the option to publish the peer review history of their article (what does this mean?). If published, this will include your full peer review and any attached files.

Reviewer #1: No

Reviewer #2: No

Reviewer #3: No

---

## [Editor Report · Acceptance letter]

18 Apr 2023

PONE-D-23-03856R1 

Surfactant Delivery via Thin Catheter in Preterm Infants: A Systematic Review and Meta-Analysis 

Dear Dr. Yeung:

I'm pleased to inform you that your manuscript has been deemed suitable for publication in PLOS ONE. Congratulations! Your manuscript is now with our production department. 

Kind regards, 

on behalf of

Prof. Harald Ehrhardt 

Academic Editor

PLOS ONE